# Relativistic Ionization of Hydrogen Atoms by Positron Impact

**Amal Chahboune, Bouzid Manaut \*, Elmostafa Hrour and Souad Taj**

Faculty Polydisciplinary, Interdisciplinary Laboratory for Research in Science and Technology, P.O. Box 523, Sultan Moulay Slimane University, Béni Mellal 23000, Morocco; a.chahboune@yahoo.fr (A.C.); e.hrour@usms.ma (E.H.); s.taj@usms.ma (S.T.)

**\*** Correspondence: b.manaut@usms.ma

**Abstract:** Relativistic triple differential cross-sections (TDCS) for ionization of hydrogen atoms by positron impact have been calculated in the symmetric coplanar geometry. We have used Dirac wave functions to describe free electron's and positron's sates. The relativistic formalism is examined by taking the non relativistic limit. Present results are compared with those for the corresponding electron-impact case. In the first Born approximation, we found that the TDCS for positron impact ionization exceeds that for electron impact for all energies in accordance with the result obtained by several other theories.

**Keywords:** positron; ionization hydrogen atom; QED calculations; relativistic scattering theory

## 1. Introduction

Impact ionization processes of atoms by charged particles are important for understanding of collision mechanisms and atomic structure. Particular attention has recently been given to positron impact ionization of atoms and molecules (Positron Physics—M. Charlton, J. Humberston (Cambridge, 2001)) [1]. Many theoretical and experimental works have been carried out for these processes [2–10].

During the past few years, the theoretical study of positron-atom ionization collisions on differential cross sections has become increasingly interesting for relativistic [11] as well as non-relativistic [12–15] energies. The ionization process of a positron with a hydrogen atom attracts much attention as one of the simplest three-body Coulomb systems. The theoretical triple differential cross section of low energy positron impact ionization of atomic hydrogen has been extensively discussed in the literature [16–19]. For the relativistic domain, a large number of theoretical investigations have been made on the TDCS for ionization of atomic hydrogen by electrons [20–22]. Using partial wave functions, a complete non relativistic formulation of the impact-ionization processes has been presented by Huang [23]. In this work we shall undertake a theoretical study of the ionization processes in the $e^+ - H$ system in the high-energy region by using the relativistic plane wave Born approximation (RPWBA). The paper is organized as follows: in Section 2 we will present the general theory of positron-impact ionization in the Relativistic Plane Wave Born approximation and we briefly describe the method used for the triple differential cross section calculations. Consistency of the theory is examined by taking the non relativistic limit, numerical results and discussions are provided in Section 3. In Section 4, we compare our results with previously reported theoretical results for the electron-impact case [20]. A summary is made in Section 5. Atomic units ($m_{e^-} = m_{e^+} = \hbar = e = 1$) will be used throughout this paper.

## 2. Theoretical Model

In positron impact ionization of hydrogen atoms, the reaction can be represented as :

$$e^+ + H \rightarrow H^+ + e^+ + e^-$$

We denote the total energy and linear momentum of the incident positron by $(E_i, p_i)$. Before the collision, the target is in its ground state with only one electron. After the collision the ion is deprived of its electron and becomes a bare nucleus. The scattered positron and ejected electron , are described by $(E_f, p_f)$ and $(E_B, p_B)$, respectively. The energies $E_f$ and $E_B$ are related by energy conservation according to :

$$E_f = E_i - E_B + \xi_b$$

where $\xi_b$ is the binding energy of electron in the ground state.

Triple differential cross section (TDCS) for the ionization of the hydrogen atom by positron impact is related to the probability that the two outgoing scattered positron and ejected electron with energies of $E_f$ and $E_B$ will be found in solid angles $d\Omega_f$ and $d\Omega_B$ after the ionization. This type of cross section determines all kinematics of particles involved in the ionization processes. Triple differential cross section is obtained from the square of the transition *S*-matrix element, multiplied by a factor which includes the momenta of the incident and scattered positron and ejected electron. The *S*-matrix element of ionization process can be written as:

$$S_{fi} = \frac{i}{c} \int_{-\infty}^{+\infty} dx^0 \langle \psi_{p_f}^p(x_1)\phi_f(x_2)|V_d|\psi_{p_i}^p(x_1)\phi_i(x_2)\rangle \tag{1}$$

where $\psi_{p_i}^p(x_1)$ and $\psi_{p_f}^p(x_1)$ are wave functions describing, respectively, incident and scattered positron, $\phi_f(x_2)$ is the wave function of the ejected electron, while $\phi_i(x_2)$ describes the ground state of the hydrogen atom.

The potential $V_d$ represents the coulomb interaction between the incoming positron and the target and is written as:

$$V_d = \frac{1}{|\mathbf{r_1}|} - \frac{1}{|\mathbf{r_1} - \mathbf{r_2}|} \tag{2}$$

where $\mathbf{r_1}$ and $\mathbf{r_2}$ are respectively, the position vectors of the incident positron and the target electron. The nucleus is taken to be infinitely heavy and fixed at the origin. Also as the particles involved in the final state are not identical we do not include exchange contributions to the TDCS.

In the RPWBA model the incident and scattered positrons and ejected electron are described by Dirac plane waves. This model has been used before to study relativistic electron impact ionization for different kinematic conditions by Y. Attaourti and S. Taj [20].

Passage from the wave function of the electron to that of the positron is obtained by the CPT transforms [24] that we apply to the Dirac equation [25], (C: Charge Transformation, P: Parity Transformation, T: Time reversion). In order to explain the particularity of the T transformation, Stuckelberg (1941), then Feynman (1948), proposed an interpretation of the T transformation based on "The idea is that a negative energy solution describes a positive-energy particle which propagates backward in time or, equivalently, a positive energy antiparticle propagating forward in time" Negative-energy particle solutions going backward in time describe positive-energy antiparticles going forward in time [26]. However the T transformation seems to imply that a positron having positive energy is treated as an electron having negative energy and propagating backward in time [27–30].

For the incident and scattered positrons, we use a free Dirac solution normalized to the volume V:

$$\psi_{p_i}^p(x_1) = \frac{v(p_f, s_f)}{\sqrt{2E_f V}} \exp^{ip_f.x_1} \tag{3}$$

$$\psi_{p_f}^p(x_1) = \frac{v(p_i, s_i)}{\sqrt{2E_i V}} \exp^{ip_i \cdot x_1} \tag{4}$$

For the ejected electron, we use a free Dirac solution normalized to the volume V:

$$\phi_f(x_2) = \psi_{p_B}(x_2) = \frac{u(p_B, s_B)}{\sqrt{2E_B V}} \exp^{-ip_B \cdot x_2} \tag{5}$$

The $v(p, s)$ and $u(p, s)$ are respectively the positron and electron bi-spinor. The hydrogen atom target is described by the relativistic wave function of atomic hydrogen in its ground state.

$$\phi_i(x_2) = \psi_t(t, r_2) = \exp^{-i\xi_b t} \psi_t(r_2) \tag{6}$$

The theoretical formalism has been developed in RPWBA and in this approximation the triple differential cross section (TDCS) has been expressed in terms of a product of kinematical factors, sums over the spins and Fourier transforms of the relativistic atomic hydrogen wave functions:

$$\begin{aligned}
\frac{d\overline{\sigma}}{dE_B d\Omega_B d\Omega_i} &= \frac{|\mathbf{p}_i||\mathbf{p}_B|}{2|\mathbf{p}_f|c^6\Delta^4}\left(\frac{1}{2}\sum_{s_i,s_f}|\overline{v}(p_i,s_i)\gamma^0 v(p_f,s_f)|^2\right)\sum_{s_B}|\overline{u}(p_B,s_B)\gamma^0|^2 \\
&\quad \times |\Phi_{1,1/2,1/2}(\Delta - \mathbf{p}_B) - \Phi_{1,1/2,1/2}(-\mathbf{p}_B)|^2
\end{aligned} \tag{7}$$

All bold symbols are vectors, $\Delta = \mathbf{p}_i - \mathbf{p}_f$ is the momentum transfer vector and functions $\Phi_{1,1/2,1/2}$ are Fourier transforms of the relativistic atomic hydrogen wave functions. Using a standard calculation in QED, spinorial parts reduce to traces in Dirac algebra such that:

$$\sum_{s_i,s_f}|\overline{v}(p_i,s_i)\gamma^0 v(p_f,s_f)|^2 = Tr[\gamma^0(c\slashed{p}_f - c^2)\gamma^0(c\slashed{p}_i - c^2)], \tag{8}$$

$$\sum_{s_B}|\overline{u}(p_B,s_B)\gamma^0|^2 = Tr[\gamma^0(c\slashed{p}_B + c^2)]. \tag{9}$$

TDCS in Equation (7) is to be compared with the corresponding one in the Non Relativistic Plane Wave Born Approximation (NRPWBA) and Non Relativistic Coulomb Born Approximation (NRCBA) :

$$\frac{d\overline{\sigma}^{(NRPWBA)}}{dE_B d\Omega_B d\Omega_i} = \frac{2^7}{(2\pi)^2}\frac{|\mathbf{p}_i||\mathbf{p}_B|}{|\mathbf{p}_f|}\frac{1}{|\Delta|^4}\left\{\frac{1}{\mathbf{q}^2+1} - \frac{1}{\mathbf{p}_B^2+1}\right\}^2, \tag{10}$$

with $\mathbf{q} = \Delta - \mathbf{p}_B$.

$$\frac{d\overline{\sigma}^{(NRCBA)}}{dE_B d\Omega_B d\Omega_i} = \frac{|\mathbf{p}_i||\mathbf{p}_B|}{|\mathbf{p}_f|}|f_{ion}^{CBA}|^2, \tag{11}$$

where $f_{ion}^{CBA}$ is the first Coulomb-Born amplitude corresponding to the ionization of atomic hydrogen by positron impact

$$f_{ion}^{CBA} = \frac{2}{|\Delta|^2}M_{1s}(\Delta, \mathbf{p}_B), \tag{12}$$

$$M_{1s}(\Delta, \mathbf{p}_B) = \frac{e^{\pi/2p_B}}{2\sqrt{2}\pi^2}\Gamma(1 - \frac{i}{p_B})\left((-\frac{dI(q)}{d\lambda})_{\lambda=1} - (-\frac{dI(-p_B)}{d\lambda})_{\lambda=1}\right), \tag{13}$$

$$I(q) = \int e^{-\lambda r}\frac{e^{i\mathbf{q}\mathbf{r}}}{r} \quad 1F1(\frac{i}{p_B}, 1, i(p_B r + \mathbf{p}_B \mathbf{r}))d\mathbf{r}. \tag{14}$$

## 3. Results and Discussion

### 3.1. Testing of the Model in Non-Relativistic Limit

We develop an exact relativistic model, in the first Born approximation, to study positron impact ionization of atomic hydrogen at high energies. In this section we test the RPWBA model proposed above by calculating relativistic TDCS for ionization by positron in a coplanar symmetric geometry. To the best of our knowledge, there is no relativistic experimental data available for comparison. For consistency checks, we simply compare our results with NRPWBA and NRCBA in the non-relativistic limit ($\gamma = 1/\sqrt{1 - \frac{v^2}{c^2}} = 1.0053$, kinetic energy ~2700 eV). We present here triple differential cross-section (TDCS) for ionization processes in $e^+ - H$ system with respect to the ejected angles $\theta_B$ for different high incident energies. The geometry chosen is $\varphi_i = \varphi_f = 0°$, $\theta_i = 0°$ and the scattering angle $\theta_f = 45°$ for the incident and scattered positron while for the ejected electron $\varphi_B = 180°$ and angle $\theta_B$ varies from $30°$ to $60°$.

Figures 1 and 2 compare the three TDCSs (RPWBA, NRPWBA and NRCBA) in the non-relativistic limit ($\gamma = 1.0053$) for positron impact ionization of H atom at (1) $T_i = 2700$ eV, $T_B = 1249.5$ eV, $\theta_f = 45°$ and (2) $T_i = 2700$ eV, $T_B = 1349.5$ eV, $\theta_f = 45°$ respectively. They are plotted versus the angle $\theta_B$ of the ejected electron.

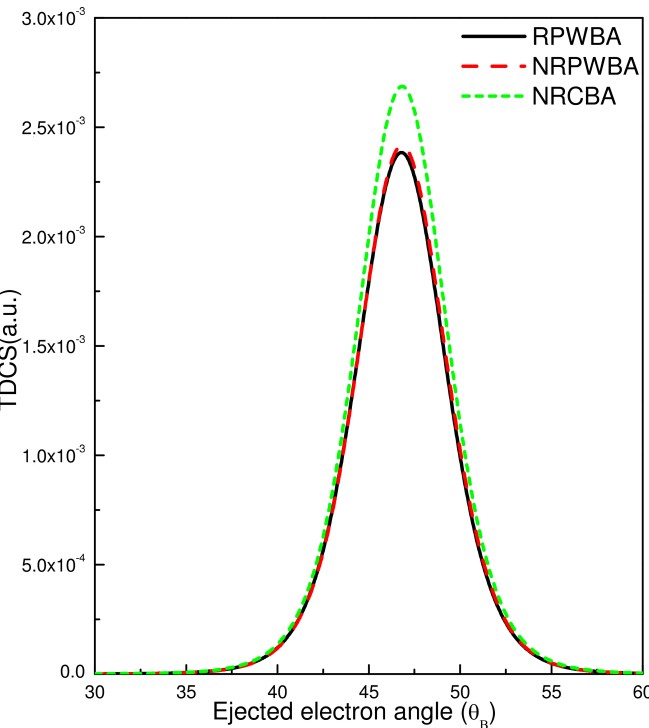

**Figure 1.** The incident positron kinetic energy is $T_i = 2700$ eV, the ejected electron kinetic energy is $T_B = 1249.5$ eV and geometric parameters are $\varphi_i = \varphi_f = 0°$, $\varphi_B = 180°$, $\theta_i = 0°$ and $\theta_f = 45°$.

From Figure 1, it can be seen that there is no difference at all between the TDCSs (RPWBA and NRPWBA) in the non-relativistic limit, whereas NRCBA gives a higher TDCS due to the fact that the ejected electron still feels Coulomb effect of the residual ion in this description. From Figure 2, we see that if we increase the ejected electron energy from 1249.5 eV to 1349.5 eV, there is a very good agreement between the three TDCSs and they give the same result. At high initial energy and for geometry with equal energy sharing by the two outgoing particles, the electron interact only weakly with the residual ion and can be described either by a plane wave or by a more general Coulomb wave

function. Three TDCSs are all peaked around $\theta_B = 45°$, for this angle, we have TDCS(NRPWBA) $\approx$ TDCS(NRCBA) $\approx$ TDCS(RPWBA) = $4.63 \times 10^{-3}$ a.u.

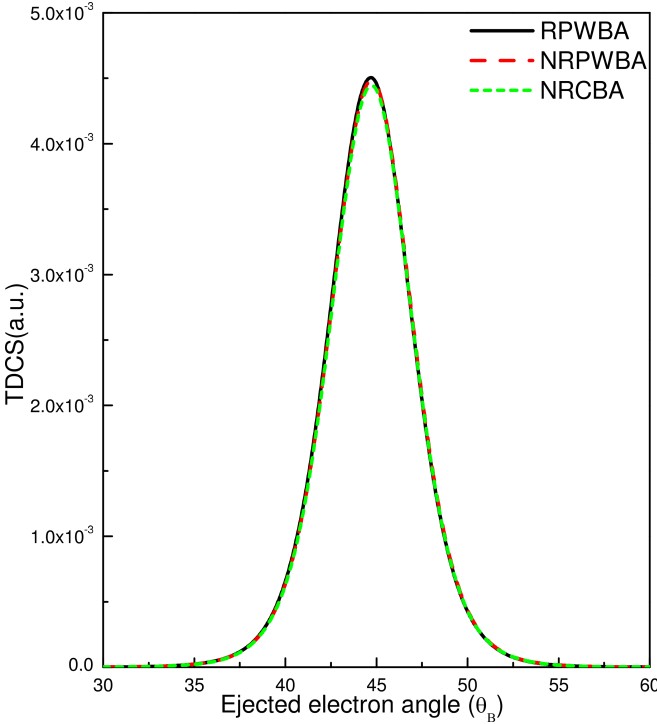

**Figure 2.** The incident positron kinetic energy is $T_i = 2700$ eV, the ejected electron kinetic energy is $T_B = 1349.5$ eV and geometric parameters are $\varphi_i = \varphi_f = 0°$, $\varphi_B = 180°$, $\theta_i = 0°$ and $\theta_f = 45°$.

### 3.2. Relativistic Regime

We present in Figures 3 and 4, for incident energies 510.999 keV and 2043.996 keV, respectively, the variation of the triple differential cross section (TDCS) of the ionization of H in terms of the ejection angle $\theta_B$ for the scattering angles 45°. We note that triple differential cross section decreases with increase of the energy of the incident positron which is the usual behavior in charged particle-impact ionization of an atom. There is a shift of the maximum of the TDCS towards smaller values than $\theta_B = 45°$.

In Table 1, we have presented the maximum values of the TDCS and peak's position results for the positron-impact ionization of hydrogen for five different incident energies with fixed scattering angle (45°).

**Table 1.** The maximum values of the triple differential cross-sections (TDCS) and the peak position for the positron-impact ionization of Hydrogen for five incident positron energies.

| $\gamma$ | Positron triple differential cross-sections (TDCS) (a.u) | $\theta_B$ (degree) |
|----------|---------------------------------------------------------|---------------------|
| 1.0053   | $4.63 \times 10^{-3}$                                   | 45                  |
| 2        | $1.29 \times 10^{-15}$                                  | 40                  |
| 3        | $4.10 \times 10^{-19}$                                  | 37                  |
| 4        | $5.03 \times 10^{-21}$                                  | 36                  |
| 5        | $2.59 \times 10^{-22}$                                  | 35                  |

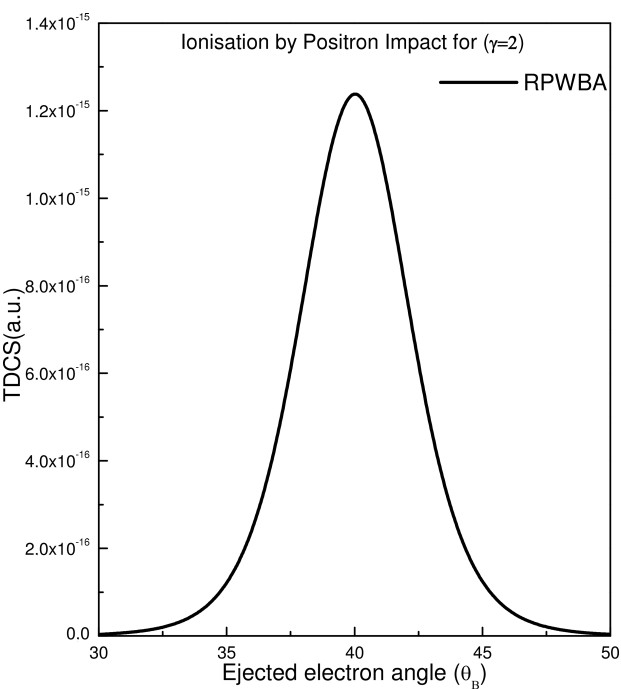

**Figure 3.** Same as in Figure 1 but for $T_i = 510.999$ keV and $T_B = 255499$ eV.

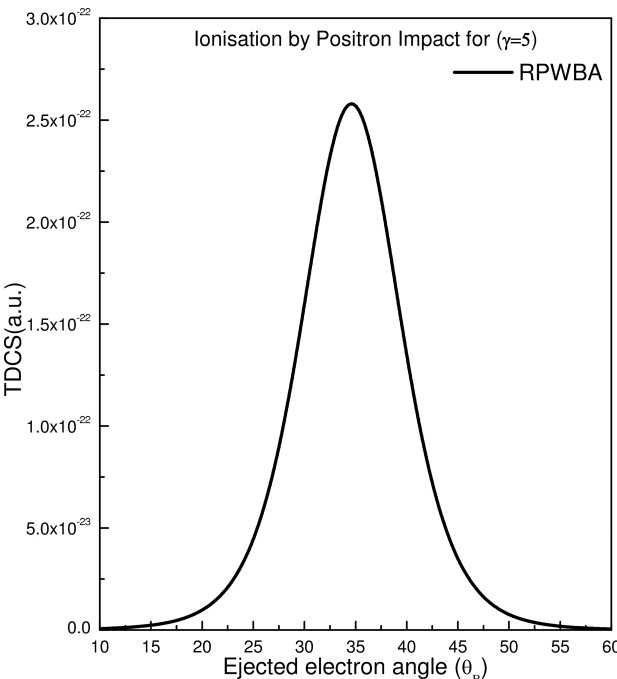

**Figure 4.** Same as in Figure 1 but for $T_i = 2043.996$ keV and $T_B = 1021.9975$ keV.

## 4. Comparison of Positron and Electron Results

It is interesting to see the effect on the triple-differential cross sections (TDCS) when the incident positron is replaced by an electron. Electron-impact ionization of hydrogen atoms has been studied theoretically within the relativistic plane wave Born approximation (RPWBA) by Y. Attaourti and S. Taj [20].

Figures 5 and 6 present a comparison of the TDCS data for positron and electron-impact ionization of H. The RPWBA electron data are in excellent agreement with the theoretical TDCS of [20]. For relativistic parameters ($\gamma$ = 1.0053 and $\gamma$ = 2), the positions of the peaks for incident electrons are exactly the same as in the positron impact case. Our TDCS positron reveals considerable differences in size compared to the electron impact ionization case under the same geometry. The peak corresponding to positron-impact is distinctly larger than the peak corresponding to electron-impact. At 2700 eV incident energy, the positron TDCS is larger by about a factor of 2.

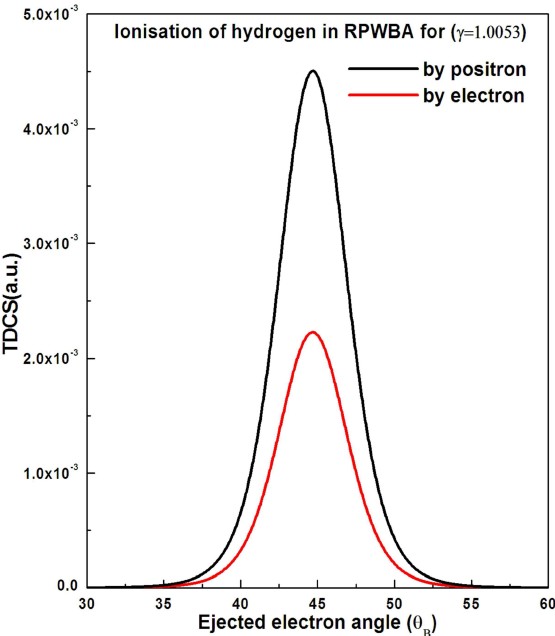

**Figure 5.** The incident kinetic energy is $T_i = 2700$ eV, the ejected electron kinetic energy is $T_B = 1349.5$ eV and geometric parameters $\varphi_i = \varphi_f = 0°$, $\varphi_B = 180°$, $\theta_i = 0°$ and $\theta_f = 45°$.

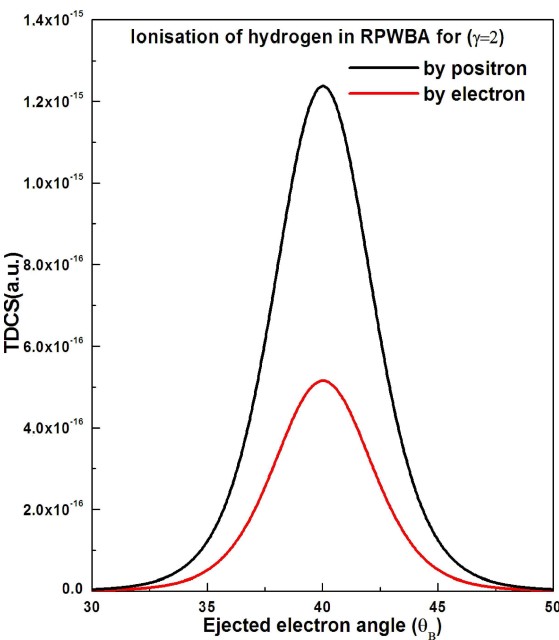

**Figure 6.** The incident kinetic energy is $T_i = 510.999$ keV, ejected electron kinetic energy is $T_B = 255.4995$ keV and geometric parameters $\varphi_i = \varphi_f = 0°$, $\varphi_B = 180°$, $\theta_i = 0°$ and $\theta_f = 45°$.

In Table 2, we have presented the maximum values of the TDCS for electron and positron impact and the values of the positron-to-electron peak ratio for specific values of incident energies with fixed scattering angle ($45°$).

The ratio of the TDCS maxima :

$$Q = max\left(\frac{d\overline{\sigma}}{dE_B d\Omega_B d\Omega_i}\right)^{positron} / max\left(\frac{d\overline{\sigma}}{dE_B d\Omega_B d\Omega_f}\right)^{electron}$$

**Table 2.** The maximum values of TDCS for electron and positron impact and values of the positron-to-electron peak ratio for specific values of incident energies.

| $\gamma$ | Positron TDCS (a.u) | Electron TDCS (a.u) | Q |
|---|---|---|---|
| 1.0053 | $4.63 \times 10^{-3}$ | $2.29 \times 10^{-3}$ | 2.02 |
| 2 | $1.29 \times 10^{-15}$ | $5.38 \times 10^{-16}$ | 2.39 |
| 3 | $4.10 \times 10^{-19}$ | $1.54 \times 10^{-19}$ | 2.66 |
| 4 | $5.03 \times 10^{-21}$ | $1.76 \times 10^{-21}$ | 2.85 |
| 5 | $2.59 \times 10^{-22}$ | $8.36 \times 10^{-23}$ | 3.09 |

The most noticeable feature in our table is the size difference between the positron and electron TDCS. The difference is very large at high impact energies and becomes relatively large with increasing energy.

Comparison of the theoretical positron and electron impact data shows that the TDCS even at relativistic energies are significantly larger for positron impact, in accordance with theoretical predictions [31] and experimental [12]. In the non relativistic case, this difference is explained qualitatively by (Brauner *et al.* (1989) [2], Joachain and Piraux (1986) [32]). For positron impact the target electron is attracted toward the positron as it moves past the target atom and this enhances the cross section. In contrast, for electron impact the electron is pushed away and cross section should decrease. This explanation of the observed changes is based on the changes in the electrostatic interactions in the ionization system. Changing the sign of the projectile charge from $+e$ to $-e$ changes attractive forces to repulsive and visa versa. This influences the scattered projectile and ejected electron kinematics, and hence the triple differentials cross section. On the one hand, this influences the impact parameter and the impact energy slightly depending on whether the projectile sees the attractive or repulsive force due to the target nucleus and its electron cloud. On the other hand, the post-collision kinematics are influenced due to the reversal of the post-collision force between the scattered projectile and ejected electron.

It could be possible to carry out similar studies for protons and anti-protons in a near future to acquire a thorough knowledge of charge and mass effects on the ionization process of the hydrogen atoms.

## 5. Conclusions

The three-body dynamics of the hydrogen ionization by positron impact have been investigated. Relativistic triple differential Cross section at several levels have been evaluated within the relativistic model (RPWBA) in the first Born approximation. The consistency of our theory is checked by taking the non relativistic limit. Fairly good agreement has been found between RPWBA, NRPWBA and NRCBA results when the outgoing particles have the same energies. We have also provided a comparison of our positron and electron impact ionization results and we can conclude that, for all energies, TDCS for positron impact is always greater in magnitude than for the electron impact.

**Author Contributions:** The authors have contributed equally to this work.

**Conflicts of Interest:** The authors declare no conflict of interest.

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
