# Peer review of "Relativistic Ionization of Hydrogen Atoms by Positron Impact"

_atoms, doi:10.3390/atoms4010010_

Reviewer 1 Report

In the paper, the authors report the theoretical calculation of triple differential cross sections of positron ionization of hydrogen atom at relativistic regime to understand the collision dynamics. This research field, concerning the three-body dynamics, is known to be a well-studied area. The main point in this work is to theoretically study relativistic ionization. However, the authors did not show further supports to their calculations, namely, by comparing the present calculations with experiments or other theoretical method (from other groups). The following issues need to be addressed:

1)      In earlier publications [Physics Reports 315 (1999) 409 and related references], a fully relativistic distorted-wave Born approximation (rDWBA) has been developed to describe the relativistic ionization processes. The authors did not mention this method for discussions in the manuscript. The RPWBA method in the manuscript should be compared with the previous rDWBA  theory. At least the authors should mention the differences of the two methods and how far the RPWBA can be compared to rDWBA?

2)      Again in earlier publications [Physics Reports 315 (1999) 409 and related references], there are existing experiments of triple differential cross sections for relativistic K-shell ionization of Ag, Au, and so on. The authors did not mention these experimental works in the manuscript. It would be nice if the RPWBA method in the manuscript can be compared with the experiments.

3)      The authors have not carefully proofread this manuscript. Evidences of this includes (i) some errors in English typing, e.g. line 27 and line 119: it should be ‘plane wave …’ instead of ‘plan wave …’. Line between 33 and 34: it should be ‘The scattered positron and ejected electron …’ instead of ‘The scattered electron and ejected positron …’. (ii) There are repeated Bibliography like ref. 11 and ref. 31.

Author Response

%%%%%%%%%%%%%%%%%%%%%%%%%%%%%%%%%%%%%%%%%%%%%%

         Answer to the first referee’s report of the Manuscript ID atoms-114670

%%%%%%%%%%%%%%%%%%%%%%%%%%%%%%%%%%%%%%%%%%%%%%

In the paper, the authors report the theoretical calculation of triple differential cross sections of positron ionization of hydrogen atom at relativistic regime to understand the collision dynamics. This research field, concerning the three-body dynamics, is known to be a well-studied area. The main point in this work is to theoretically study relativistic ionization.  However, the authors did not show further supports to their calculations, namely, by comparing the present calculations with  experiments or other theoretical method (from other groups). The following issues need to be addressed:

  ------------------------------- 

 Answer of authors 

 ------------------------------- 

First of all, we would like to thank you very much because you have seen at first glance the importance of this work using QED  formalism. 

1) Earlier publications [Physics Reports 315 (1999) 409 and related references], a fully relativistic distorted-wave  Born approximation (rDWBA) has been developed to describe the relativistic ionization processes. The authors did not mention   this method for discussions in the manuscript. The RPWBA method in the manuscript should be compared with the previous rDWBA  theory. At least the authors should mention the differences of the two methods and how far the RPWBA can be compared to rDWBA?

 ------------------------------- 

1) Answer of authors 

------------------------------- 

 The review article by W Nakel and C. T. Whelan  you refer to do indeed tackle the relativistic regime but at the expense of a huge amount of numerical work. So, we think that presenting for the first time a work that incorporates fully analytical results and QED formalism is interesting to the scientific community. The work of W Nakel and C. T. Whelan is devoted to study heavy atoms but Our work is aimed to study analytically the hydrogen atom, we believe that such a comparison is impossible. Another difference that makes comparison difficult is that the work of W Nakel and C. T. Whelan uses coplanar asymmetric geometry while in our work we 

use coplanar symmetric geometry.

---------------------------------------------------------------------------------------------------------------------------------

2) Again in earlier publications [Physics Reports 315 (1999) 409 and related references], there are existing experiments of triple differential cross sections for relativistic K-shell ionization of Ag, Au, and so on. The authors did not mention these experimental works in the manuscript. It would be nice if the RPWBA method in the manuscript can be compared with the experiments.

------------------------------- 

2) Answer of authors 

------------------------------- 

We are fully aware that the mainstream trend is to study heavy atoms. But some fundamental processes have not until now been investigated fully and analytically and it is our very humble belief that such a fundamental work must be presented to the international scientific community. We are not in a position to compare with these works since they are devoted to heavy atoms.

----------------------------------------------------------------------------------------------------------------------------------------

3)  The authors have not carefully proofread this manuscript. Evidences of this includes 

(i) some errors in English typing, e.g. line 27 and line 119: it should be ‘plane wave …’ instead of ‘plan wave …’.  Line between 33 and 34: it should be ‘The scattered positron and ejected electron …’ instead of ‘The scattered electron and ejected positron …’.

(ii) There are repeated Bibliography like ref. 11 and ref. 31.

------------------------------- 

3) Answer of authors 

------------------------------- 

Points (i) and (ii) have  been  corrected and changed  following  to  the  referee’s amendments.

------------------------------------------------------------------------------------------------------------------------------------------

We hope that we have given convincing answers to your remarks as they were very sound and relevant. Again, we appreciate all your insightful comments. 

Thank you for taking the time and energy to help us improve the paper.

Reviewer 2 Report

This manuscript reports new TDCS calculations for relativistic positron and electron impact on atomic hydrogen.  Although no experimental data are available for comparison, the method is well explained and reported to agree with other electron impact calculations.  In my opinion, the primary contribution of this work is the prediction that even at relativistic velocities, the TDCS for positron impact is larger than for electron impact and that the difference increases with impact energy.  In spite of the lack of any experimental, or, as is claimed, any theoretical, verification of this or any physical interpretation as to why this should occur even at very high velocities, I believe that work deserves publication since it should incite additional theoretical attempts to verify these predictions.  Also, hopefully sometime in the future, TDCS studies for positron impact will advance to the stage where the present predictions can be tested.  I find no major problems with the manuscript, either in content or presentation.  As for minor problems, there are numerous English grammatical errors which I have attempted to list in order to facilitate publication.  In addition, I do believe that several figures could be combined in order to minimize journal space.  These are listed below.

My overall recommendation is that after the listed items are addressed the manuscript be accepted for publication without further review.

·         Title:  atoms, not atom

·         Line 3: states is misspelled.   Insert “The” before “relativistic”.  Is, not are

·         Line 5: insert “the” before “first” and before “TDCS”

·         Line 14-15: change “a such process” to “these processes”

·         Line 19: systems  (plural)

·         Line 25: don’t capitalize relativistic

·         Line 27: Plane is misspelled

·         Line 29: insert “the” before “non”

·         Line 31: insert “for the” before “electron-impact”

·         Line 33: delete comma before “are described”

·         Line 34: insert “the” before “ground state”

·         Line 35: either “atoms”  (plural) or insert “the” before “hydrogen”

·         Line 43: insert “the” before “ground state”

·         Line 44: insert “the” before “hydrogen”

·         Line 47: “include” would be a better choice for “take”

·         Line 48: insert “the” before RPWBA   and delete the comma

·         Line 51: insert “the” before “wave” and before “electron” and before “positron”

·         Line 52: insert “the” before “Dirac”

·         Line 53: “explain” not explaining

·         Lines 58-59: insert “a” before “positron” and “an” before “electron”

·         Lines 63 and 64: insert “The” at start of sentence and before “relativistic”

·         Lines 67-68:  it should be stated that the bold symbols are vectors

·         Line 69: insert either “the” or “a” before “standard”

·         Line 70: in the equation, check cpi and cpB in as they view strange in my pdf copy

·         Line 71: insert “the” before “Non”

·         Line 73: in equation 7, q needs to be defined

·         Line 81: insert “the” before “first”

·         Line 85: either “a consistency check” or “consistency checks”

·         Line 86: please check with journal whether a comma or period should be used in 1,0053

·         Line 86: change “have presented” to “present”

·         Lines 88-89: insert “the” before “Geometry” , “scattering”, “incident” and “ejected”

·         Lines 91-92: I suggest deleting the first sentence of this paragraph since, with varying degrees of accuracy, all 3 wave functions can be used, as is shown.

·         Line 92: delete “a”

·         Line 93: insert “the” before “non-“

·         Line 97: insert “the” before “non” and “ejected”

·         Line 98: insert “the” before “residual”

·         Line 102: insert “a” before “plane”

·         Line 103: function (singular)

·         Lines 99 and 104 and Table I: again, check with journal whether a comma or period should be used in the numbers

·         Captions for figures 1 and 2: insert “are” before “geometric”   also check whether commas or periods should be used in the numbers

·         Line 106 and figures 3 and 4: giving the energies in keV and rounding to the keV value is preferable.  Also, the shift in angle would be more apparent to the reader if identical angular scales were used.  Plus, I recommend combining the two figures and using some sort of velocity scaling on the vertical axis to illustrate the falloff in magnitude.

·         Line 108: don’t capitalize “triple”

·         Line 112 and caption for Table I:  insert “the” before “TDCS: and “peaks” and change “peak’s” to “peak” without italics

·         Line 115: delete “for positron impact” at end of sentence.

·         Table I:  check journal standards for using comma or period plus period to signify times power of 10

·         Line 123: again, check journal standards on numbers   and “electrons” plural

·         Line 124: “Our positron TDCS reveals”

·         Line 125: insert “the” before “electron”

·         Line 127: suggestion, replace “different in magnitude….electron’s one” by “larger”

·         Caption, fig 5 and 6:  check number notation.  Also, round the numbers in Fig 6 to whole keV units.

·         Paragraph, lines 121-128: both fig. 5 an 6 are mentioned but only 5 is discussed.  I suggest combining the figures to parts a and b and modifying the text as needed. 

·         Line 129: insert “the” before “TDCS”

·         Table 2: number notation

·         Line 134: replace “by” with “with”

·         Lines 135-137: delete this sentence as it says exactly the same as the previous sentence and Table 2.

·         Line 138: insert “the” before “TDCS”   also inserting “even at relativistic energies” would strengthen this intent of this work

·         Line 139: reference 30 should be replaced with a theoretical reference or the text changed to not imply that ref 30 is a theoretical work

·         Line 140: delete “one” and insert “the” before “non” plus change “that this would enhance” to “this enhances”  Also lines 141-143 are probably meant to be a single sentence.  Please correct.

·         Line 143: change “The” to “This”

·         Line 144: change “should be” to “is”

·         Line 148: insert “the” before “attractive”

·         Line 149:  capitalize “on”

·         Paragraph 138-153: Arguments for why the positron cross section is larger at low energies are given but no discussion as to if they would, or would not, apply at high energies is given.  It could be argued that in very fast collisions the electron cloud has no time to respond so some other explanation is needed.  Please add additional comments.

·         Line 155: change “impact of positron has” to “positron impact have”

·         Line 157: insert “the” before “first”

·         Line 160: delete “of the electron” and insert “and electron” before “impact”

·         Line 162: replace “these of the” with “for”

Author Response

%%%%%%%%%%%%%%%%%%%%%%%%%%%%%%%%%%%%%%%%%%%%%%

         Answer to the second referee’s report of the Manuscript ID atoms-114670

%%%%%%%%%%%%%%%%%%%%%%%%%%%%%%%%%%%%%%%%%%%%%%

This manuscript reports new TDCS calculations for relativistic positron and electron impact on atomic hydrogen. Although no experimental data are available for comparison, the method is well explained and reported to agree with other electron impact calculations.  In my opinion, the primary contribution of this work is the prediction that even at relativistic velocities, the TDCS for positron impact is larger than for electron impact and that the difference increases with impact energy.  In spite of the lack of any experimental, or, as is claimed, any theoretical, verification of this or any physical interpretation as to why this should occur even at very high velocities, I believe that work deserves publication since it should incite additional theoretical attempts

to verify these predictions.  Also, hopefully sometime in the future, TDCS studies for positron impact will advance to the stage where the present predictions can be tested.  I find no major problems with the manuscript, either in content or presentation.  As for minor problems, there are numerous English grammatical errors which I have attempted to list in order to facilitate publication.  In addition, I do believe that several figures could be combined in order to minimize journal space.  These are listed below. My overall recommendation is that after the listed items are addressed the manuscript be accepted for  publication without further review. 

 ------------------------------- 

 Answer of authors 

 ------------------------------- 

- First of all, we would like to thank you very much for this relevant and detailed report.

- As you requested, we have made all necessary changes in our manuscript.  A major revision of   the paper has been carried out to take all the comments, suggestions and english corrections     into account. We believe that the paper has been significantly improved. 

- We  are  very  grateful  to  you  for  your  sound  and  relevant  corrections.  

Reviewer 3 Report

The article presents triple differential cross sections (tdcs) calculations of hydrogen ionization by positron and electron impact, in the relativistic and non-relativistic limit, using relativistic plane-wave Born approximation and non relativistic approximations. The results confirm predictions of other theories. Explanation of the differences between the peaks corresponding to the different projectiles is given. The work and discussion are very short, but still interesting.

I recommend it for publication after the authors answer the questions and make the corrections described below:

Page 1:
* Line 7: it is referenced to the recent theories given in [2] and [3] which are from years 1994 and 2002 respectively. Maybe the word "recent" is not adequate here.
* Line 19: "Coulomb system" -> "Coulomb systems"

Page 2:
* Line 36: "describes the probability": actually it is the flux ratio, and it could be divergent in some cases. I think one should be careful. Maybe the authors could use "is related to the probability" or something like that.
* Line 40: "positrons" -> "positron"
* Line 43: "positrons" -> "positron" and "ground state" -> "the ground state".

Page 3:
* unnumbered equation between eqs. (1) and (2) for Vd: in the second term of the right hand side (rhs) the quantities between the vertical bars are vectors and the notation should be different from that of the first term of the rhs, which represent a modulus.
* unnumbered equations between eqs. (6) and (7): there should be a connector between the two equations such as a comma or "and", the same for  eqs. (7) to (8) and (9) to (11).

Page 4:
* Line 80: "of model" -> "of the model"
* Line 81: "in first" -> "in the first"
* line 86: the gamma letter (the Lorentz factor) must be defined in the text.

Page 5:
* Lines 103 and 104: Maybe the identification of the used approximation for the three TDCS could be as subscript.
* The fact that as the incident energy increases, the TDCS decreases and shifts towards smaller ejection angles for positron impact is said twice between lines 108 and 115.

Page 9:
* "experimental one" -> "experimental ones"

Author Response

%%%%%%%%%%%%%%%%%%%%%%%%%%%%%%%%%%%%%%%%%%%%%%

         Answer to the third referee’s report of the Manuscript ID atoms-114670

%%%%%%%%%%%%%%%%%%%%%%%%%%%%%%%%%%%%%%%%%%%%%%

The article presents triple differential cross sections (tdcs) calculations of hydrogen ionization by positron and electron impact, in the relativistic and non-relativistic limit, using relativistic plane-wave Born approximation  and non relativistic approximations. The results confirm predictions of other theories. Explanation of the differences   between the peaks corresponding to the different projectiles is given. The work and discussion are very short, but still interesting.

I recommend it for publication after the authors answer the questions and make the corrections described below:

------------------------------- 

 Answer of authors 

 ------------------------------- 

First of all, we would like to thank you very much for this relevant report. 

--------------------------------------------------------------------------------------------------------------------------

Page 1:

* Line 7: it is referenced to the recent theories given in [2] and [3] which are from years 1994 and 2002 respectively.  Maybe the word "recent" is not adequate here.

 * Line 19: "Coulomb system" -> "Coulomb systems"

------------------------------------------- 

 Answer of authors 

------------------------------------------- 

 The changes is  made in the text.

-------------------------------------------

Page 2:

* Line 36: "describes the probability": actually it is the flux ratio, and it could be divergent in some cases.  I think one should be careful. Maybe the authors could use "is related to the probability" or something like that. 

* Line 40: "positrons" -> "positron"

* Line 43: "positrons" -> "positron" and "ground state" -> "the ground state".

------------------------------------------- 

 Answer of authors 

------------------------------------------- 

 The changes is  made in the text.

-------------------------------------------

Page 3:

* unnumbered equation between eqs. (1) and (2) for Vd: in the second term of the right hand side (rhs) the quantities between the vertical bars are vectors and the notation should be different from that of the first term of the rhs, which represent a modulus. 

* unnumbered equations between eqs. (6) and (7): there should be a connector between the two equations  such as a comma or "and", the same for  eqs. (7) to (8) and (9) to (11).

------------------------------------------- 

 Answer of authors 

------------------------------------------- 

 The changes is  made in the text.

-------------------------------------------

Page 4:

* Line 80: "of model" -> "of the model"

* Line 81: "in first" -> "in the first"

* line 86: the gamma letter (the Lorentz factor) must be defined in the text.

------------------------------------------- 

 Answer of authors 

------------------------------------------- 

 The changes is  made in the text.

-------------------------------------------

Page 5:

* Lines 103 and 104: Maybe the identification of the used approximation for the three TDCS could be as subscript.

* The fact that as the incident energy increases, the TDCS decreases and shifts towards smaller ejection angles for positron impact is said twice between lines 108 and 115.

------------------------------------------- 

 Answer of authors 

------------------------------------------- 

  The changes is  made in the text.

-------------------------------------------

Page 9:

* "experimental one" -> "experimental ones"

------------------------------------------- 

 Answer of authors 

------------------------------------------- 

  The changes is  made in the text.

-------------------------------------------

Thank you once again for your valuable comments and corrections. 

Round  2

Reviewer 1 Report

The manuscript in the present form is suggested to be accepted for publication